# ADAPTIVE PREDICTION TIME FOR SEQUENCE CLASSIFICATION

**Maksim Ryabinin, Ekaterina Lobacheva**
National Research University Higher School of Economics, Moscow, Russia
mkryabinin@edu.hse.ru, elobacheva@hse.ru

## ABSTRACT

In this paper, we propose a recurrent neural network architecture for early sequence classification, when the model is required to output a label as soon as possible with negligible decline in accuracy. Our model is capable of learning how many sequence tokens it needs to observe in order to make a prediction; moreover, the number of steps required differs for each sequence. Experiments on sequential MNIST show that the proposed architecture focuses on different sequence parts during inference, which correspond to contours of the handwritten digits. We also demonstrate the improvement in the prediction quality with a simultaneous reduction in the prefix size used, the extent of which depends on the distribution of distinct class features over time.

## 1 INTRODUCTION

The task of sequential data classification has numerous applications in the fields of natural language processing (Bijalwan et al., 2014), video (Wang & Sng, 2015) and time series (Kampouraki et al., 2009; Krollner et al., 2010) analysis. However, methods used for these problems often assume that the whole sequence is available for the model to use, which is not true, for example, in the situations when the sequence length is unknown at inference time or when lack of ability to predict early can have negative consequences. Examples of applications that impose such restrictions are the analysis of surroundings for automated driving systems (the model needs to react in a correct way to sudden environmental changes, such as pedestrians or approaching cars, as soon as possible), patient record monitoring for medical purposes, video surveillance systems and financial time series classification.

An inherent difficulty of the early prediction task is a need to balance between two contradictory objectives: on the one hand, the sequence prefix used needs to be noticeably shorter in order for the initial requirement to be satisfied; on the other hand, if the number of observations made is too small, the model may not possess enough information to give a correct answer.

Recurrent neural networks and their modifications, such as LSTM (Hochreiter & Schmidhuber, 1997) and GRU (Cho et al., 2014), are a popular approach to sequence modeling but the original formulation of these models is not capable of returning a final prediction before reaching the sequence end as there is no possibility for a network to explicitly indicate whether no further steps are required. Several methods (Wang et al., 2016; Aliakbarian et al., 2017) for early sequence classification with RNNs have been proposed previously, mainly making use of loss functions that are specifically designed to encourage returning a label early on.

Closely related to the task mentioned above is the actively researched class of models (Yeung et al., 2016; Yu et al., 2017; Campos et al., 2018) that are trained to skip timesteps in a sequence in order to reduce the amount of computation or to improve the gradient flow for training by backpropagation through time. Certain architectures from this class are also capable of making a decision to stop, yet these models are primarily aimed at solving a different task and therefore are not effective for early sequence classification.

There also have been proposed several architectures for variable computation designed for the text classification task only: in Dulac-Arnold et al. (2011), a simple approach based on approximate Q-learning is used to train an agent which can either assign a label to the current document, move to the next sentence, or stop reading; the earliness requirement is enforced by the reward function.

Another model is introduced in Shen et al. (2017), where an RNN-based agent is used to compute attention over the sequence elements several times until a satisfying answer is found, halting the loop by sampling a Bernoulli random variable from the distribution specified by a network layer at each reasoning step. The REINFORCE algorithm (Williams, 1992) is used to optimize the objective. Keyi Yu (2018) propose an architecture which includes a policy module defining the action to be made after each token: the agent may either reread the token, skip several next sequence elements or make a prediction and stop reading. The model is also trained by the policy gradient method and is directly penalized for reading too many tokens, which makes it suitable for early prediction. However, all three models described above have been designed for the sole purpose of text classification, whereas the model proposed in this paper does not depend on the type of data used.

Our approach to the early prediction task consists of a recurrent neural network with a sigmoidal halting unit that is inspired by Graves (2016), where it was used for performing multiple computation steps for a single sequence element. The unit outputs the probability for the network to stop after each step and no further steps are made when these probabilities form a valid distribution (all outputs are non-negative and sum to 1). The whole architecture is end-to-end differentiable and is directly trained to predict a label for a sequence as early as possible. Moreover, the halting scores may be interpreted as an estimate of current token importance made by the network, which allows us to draw a connection between the method proposed and the attention mechanism introduced in Bahdanau et al. (2014) and widely used for tasks dealing with sequential data.

## 2 ADAPTIVE PREDICTION TIME

Our approach is inspired by a similar architecture discussed by Graves (2016), where it is used for determining the amount of computation steps for every token. The proposed architecture consists of three main elements: a recurrent network for encoding input tokens, the sigmoidal halting unit determining the probability to stop after current input, and a classifier layer which outputs relative score for each possible sequence label.

Let $T$ be the total sequence length, $x_t$ and $s_t$ be the input and hidden state vectors of the recurrent network at time step $t$, and $\theta$ be a set of parameters of this network. At each timestep $t$ the proposed model first computes the hidden state $s_t$ and then feeds it into the sigmoidal halting unit, which is simply a fully-connected layer with sigmoid as an activation function:

$$s_t = RNN(x_t, s_{t-1}, \theta), \qquad h_t = \sigma(W_h s_t + b_h). \tag{1}$$

We use GRU (Cho et al., 2014) in our model, although use of other RNN variations is also possible.

The resulting activation is then used to obtain the halting probability of each step:

$$p_t = \begin{cases} R \text{ if } t = N, \\ h_t \text{ otherwise,} \end{cases} \tag{2}$$

where

$$N = \min\left\{t' : \sum_{t=1}^{t'} h_t \geq 1 - \varepsilon, \ T\right\}, \qquad R = 1 - \sum_{t=1}^{N-1} h_t \tag{3}$$

and $\varepsilon$ is a small constant to allow the computation to stop after a single step. $N$ is the number of the final step, which occurs either after the sum of halting scores has exceeded the threshold defined by $\varepsilon$ or after all tokens have been read and $R$ is the remainder which is used in order to impose a requirement on sum of all $p_t$ to be equal to 1. By also noting that $p_t \geq 0 \ \forall t \in \{1, \ldots, T\}$ we conclude that $p_t$ form a probability distribution over possible timesteps for a network to halt, therefore when $t$ reaches $N$, no further steps are made. Then the input vector $\hat{s}$ for the classifier layer is defined by the expectation over the obtained distribution: $\hat{s} = \sum_{t=1}^{N} p_t s_t$ and the label probabilities are estimated by feeding $\hat{s}$ into the softmax layer. In order to place constraints on the number of steps made we optimize the network weights by cross-entropy loss $L$ while also adding the remainder (which is a linear function of network outputs) with a factor $\lambda$:

$$\hat{L}(x, y) = L(x, y) + \lambda R. \tag{4}$$

Here $\lambda$ is responsible for a trade-off between classification accuracy and a number of tokens we need to classify a sequence.

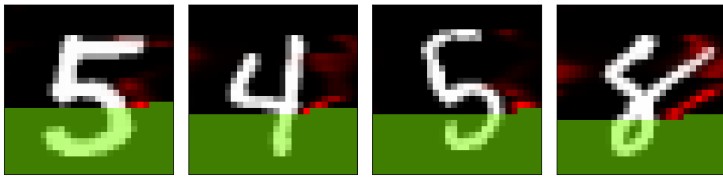

Figure 1: Visualization of halting probabilities for sequential MNIST digits. The red color denotes the $p_t$ magnitude, whereas the pixels not used by the model are marked by green color

Table 1: Results on regular and reordered sequential MNIST datasets.

| Model | Regular seqMNIST | | | Reordered seqMNIST | | |
| --- | --- | --- | --- | --- | --- | --- |
| | Test accuracy | Steps | Prefix | Test accuracy | Steps | Prefix |
| GRU | 97.4 | 784 | 784 | 11.32 | 784 | 784 |
| Skip GRU | 97.5 | **393.78** | 783.18 | 89.45 | 282.28 | 774.12 |
| APT, $\lambda = 0$ | 98.44 | 784 | 784 | - | - | - |
| APT, $\lambda = 1.5 \cdot 10^{-3}$ | **98.71** | 711.6 | 711.6 | - | - | - |
| APT, $\lambda = 10^{-2}$ | 98.09 | 593.2 | **593.2** | 92.3 | 262.4 | 262.4 |

## 3 EXPERIMENTS

First, we evaluate the proposed method on the classification of sequential MNIST dataset. In this task pictures from MNIST dataset (LeCun et al., 1998) are fed into the network as sequences of pixels, so the algorithm accesses each image row by row one pixel at a time. As baselines we use a GRU network without halting unit and a Skip-RNN model by Campos et al. (2018). We train all models for 600 epochs using RMSprop optimizer (Hinton et al.) with a learning rate of $10^{-3}$ and batch size of 128. The hidden state size of each network is 110. Gradients are clipped during training so that their norm does not exceed 1.

The results of the experiment are shown in the left part of Table 1. The proposed model is capable of classifying rather long sequences with quality on par or even better than other models while also using shorter prefixes of input data. Even though Skip GRU needs to process less pixels to make a prediction, it has to look almost through the whole sequence and therefore is not suitable for early prediction task. It is also worth noting that the Skip GRU uses approximately every second sequence element, which is possible only because of specifics of data used.

We have also found that there are different patterns in halting probabilities emitted by our model for each class, often repeating the digit contours if there are no bright pixels for the rest of the line. Examples of such patterns are shown in Figure 1. A similar behavior was observed in the model for images by Figurnov et al. (2017), which is also based on ACT Graves (2016) . This process of focusing on important parts of the sequence with further use of these scores for reweighing token representations might be compared to the concept of attention (Bahdanau et al., 2014) used for sequence modeling by recurrent neural networks. The major difference is that our approach computes the probabilities as the input is processed without any other information than the current network state instead of calculating the attention scores after the encoding phase. A possible explanation of high performance on long sequences might be the possibility for a network to remember important parts by increasing their halting probabilities.

The fact that reduction in prefix size is just approximately $25\%$ from the total sequence length may be based on the specific sequence structure: the most 'important' information about a digit is contained in the center of the image, while the pixels close to the border are rarely of any use to the network. In order to show that it is possible for our network to significantly reduce the number of tokens consumed when the sequence beginning contains valuable information, we use a modified version of the sequential MNIST dataset, where all pixels are sorted in decreasing order by their variance. As shown in the right part of Table 1, this rearrangement allows us to see that it is possible for our model to cut down the sequence length used by more than $60\%$, making even less steps than the Skip-RNN baseline.

ACKNOWLEDGMENTS

The authors would like to thank Michael Figurnov and Dmitry Vetrov for helpful input on this paper and for insightful discussions.

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
