# OpenReview forum: "Adaptive prediction time for sequence classification"
_ICLR.cc/2018/Workshop — Reject_

### Official Review · AnonReviewer2 · 2018-03-06
**Some positive results, but the paper suffers from unfortunate nomenclature and poorly justified claims.**

**Rating:** 4
**Confidence:** 4

**Review:**

The authors propose a sequence classification model which uses an attention mechanism whose structure allows the model to only use a prefix of the input. This allows them to achieve a small increase in accuracy over some baselines on the sequential MNIST dataset, using a number of operations comparable to the SkipRNN model of Campos et al.

The choice of calling h_t a 'halting probability' is somewhat confusing. At no point is the model stopped by sampling from a h_t Bernouilli, as one might expect from the name. Instead, the h_t correspond to attention weights computed sequentially as independent sigmoids until their sum reaches 1, at which point the model stops reading the input.

There are several issues with that setup. First, there is no guarantee that p_t sum to one: what happens e.g. id \epsilon = 0.2, h_1 = 0.75 and h_2 = 0.75 ? then, according to equations (1)-(3), we have p_1 + p_2 = 1.5. Secondly, R is not a function of the sequence length, and so I am not sure I understand how \lambda trades off between accuracy and computation.

In the Experiments section, the authors need to give more details about how the number of steps is counted: I assume that it is the average value of N as defined in Equation (N), but it is not absolutely clear.

Finally, claiming an improvement over Skip-RNN on the basis of a reduction in the prefix length on Reordered SeqMNIST does not make sense: as the authors note themselves, this is only possible because the specific reordering gives them access to important information from the future earlier, and does not correspond to any reality of the motivating applications mentioned in the Introduction.

---

> ### Author Response · Authors · 2018-05-10
> **Author's response**
>
> We would like to thank you for your thorough review and address some of the concerns in detail below.
>
> «The choice of calling h_t a 'halting probability' is somewhat confusing. At no point is the model stopped by sampling from a h_t Bernouilli»
> You are right about the name for h_t being rather misleading for the readers. However, as our setup is inspired by Graves (2016) paper, we tried to use the same notation to describe the model, but it is probably a good idea to add additional clarification to the paper.
>
> «First, there is no guarantee that p_t sum to one: what happens e.g. id \epsilon = 0.2, h_1 = 0.75 and h_2 = 0.75 ? then, according to equations (1)-(3), we have p_1 + p_2 = 1.5»
> Sum of all p_t is equal to one, because the last p_t is always set to complement other probabilities to 1. If h_1=0.75 and h_2=0.75, then h_1+h_2>=1-\epsilon, then N=2 and p_2=0.25, so we get p_1+p_2=1 by construction.
>
> «Secondly, R is not a function of the sequence length, and so I am not sure I understand how \lambda trades off between accuracy and computation.»
> Thank you for this observation, the explanation was indeed needed to be more thorough. The idea is that gradients of R with respect to all halting probabilities except the last equal to -1; thus by minimizing R we increase these halting probabilities and because of this the overall sum exceeds 1-\epsilon at an earlier step. The initial motivation for optimizing R comes from introducing ponder cost as the penalty term which is calculated as R+N; however, because N is piecewise constant, its gradients are discarded for the optimization process.
>
> «In the Experiments section, the authors need to give more details about how the number of steps is counted: I assume that it is the average value of N as defined in Equation (N), but it is not absolutely clear»
> As with the previous question, we will extend the description in the next version of the paper. The number of steps is calculated as the number of RNN state updates for each sequence, averaged over the test set. This equals to the average value of N for our model, the average sequence length for GRU baseline, and the average number of steps for which the state update gate was set to 1 for Skip GRU model.
>
> «Finally, claiming an improvement over Skip-RNN on the basis of a reduction in the prefix length on Reordered SeqMNIST does not make sense…this is only possible because the specific reordering gives them access to important information from the future earlier»
> We agree that the comparison between our model and Skip-RNN on the reordered sequences is unfair, as the reordered sequences have several hundred black pixels in the end, making the network forget the initial information even when the updates are rare. However, the goal of this experiment was not to show an improvement in classification quality or sequence length in comparison with other models but to show that it is possible for our network to significantly reduce the length of the prefix used depending on the location of discriminative information in the sequence. In other words, this was just a toy example to prove that our model is indeed capable of predicting much earlier than reaching the end of the sequence.

---

### Official Review · AnonReviewer3 · 2018-03-10
**useful and appropriate application of existing method, not much novelty**

**Rating:** 6
**Confidence:** 5

**Review:**

The paper is inspired by Graves (2016), which allows the model to apply RNN multiple times (# of times also learned by the model) for each input to the RNN. The paper uses the same mechanism to halt the RNN early on sequential modeling while maintaining differentiability. So it is using an existing method for a different goal. This seems to be an appropriate application and could be useful but there is not much novelty of the method. The experimental visualizations are convincing and interesting but could have been better if also used on other sequence datasets (such as language classification) than MNIST.

---

> ### Author Response · Authors · 2018-05-10
> **Author's response**
>
> Thank you for your feedback. We are going to report other experimental results on data such as video and text in the next revision of the paper.

---

### Official Review · AnonReviewer1 · 2018-03-10
**Simple solution to adaptive prediction time.**

**Rating:** 7
**Confidence:** 5

**Review:**

The paper considers the task of adaptive prediction time - or in other words learning to produce answers when enough data was observed.

The proposed solution uses the ACT (Graves 2016) computation as the stopping criterion. However, (Graves 2016) used it to determine the number of ponder steps made after every RNN transition, while in this submission the computation is done globally, to determine the overall number of transition steps.

Questions:
why does the skip GRU make fewer steps in the reordered MNIST rather than the regular MNIST? It seems that in both cases it was looking ad every second pixel. Or was this number tuned? Then you may want to indicate it in the table.

Pros:
- It is nice to see that the ACT computation, originally developed to handle sequences of a few ponder steps scales to hundreds of steps.

Cons:
- The approach seems to be limited by the data ordering - the biggest gain in time reduction is obtained when a better permutation of MNIST pixels is used. This suggests that the technique must find its own proper benchmark problems.

---

> ### Author Response · Authors · 2018-05-10
> **Author's response**
>
> Thank you for your helpful comments.
>
> Regarding the question about fewer steps for Skip GRU on reordered MNIST: as with our model, the network automatically determines how many steps to make depending on the data and attempts to skip noninformative sequence elements. Because nearly all pixels from the center of the image are moved to the beginning, the network might read most of them (possibly skipping redundant ones) as in the original dataset and then skip large image portions which are mainly black until the end of the sequence. We will add a detailed comparison with illustrations to the next revision of the paper.
>
> As for dependence of reduction in the number of steps on the data ordering, we will further explore this phenomenon in additional experiments using other datasets in order to conduct a more comprehensive study.

---

### Decision · Program_Chairs · 2018-03-20
**ICLR 2018 Workshop Acceptance Decision**

**Decision:**

Reject

**Comment:**

Based on the reviews, this paper has not been accepted for presentation at the ICLR workshop. However, the conversation and updates can continue to appear here on OpenReview.